# Combined Use of Sentinel-1 SAR and Landsat Sensors Products for Residual Soil Moisture Retrieval over Agricultural Fields in the Upper Blue Nile Basin, Ethiopia

**DOI:** 10.3390/s20113282

**Published:** 2020-06-09

**Authors:** Getachew Ayehu, Tsegaye Tadesse, Berhan Gessesse, Yibeltal Yigrem, Assefa M. Melesse

**Affiliations:** 1Remote Sensing Research and Development Department, Entoto Observatory & Research Center, Ethiopian Space Science and Technology Institute, Addis Ababa 33679, Ethiopia; getachewt@essti.gov.et (G.A.); berhang@gmail.com (B.G.); 2Institute of Land Administration, Bahir Dar University, Bahir Dar 79, Ethiopia; 3National Drought Mitigation Center, University of Nebraska-Lincoln, Lincoln, NZ 830988, USA; 4Department of Geography and Environmental Studies, Bahir Dar University, Bahir Dar 79, Ethiopia; yibeltala@gmail.com; 5Department of Earth and Environment, Florida International University, Miami, FL 33199, USA; melessea@fiu.edu

**Keywords:** synthetic aperture radar, Sentinel-1, Landsat, backscattering, ANN, residual soil moisture, surface roughness

## Abstract

The objective of this paper is to investigate the potential of sentinel-1 SAR sensor products and the contribution of soil roughness parameters to estimate volumetric residual soil moisture (RSM) in the Upper Blue Nile (UBN) basin, Ethiopia. The backscatter contribution of crop residue water content was estimated using Landsat sensor product and the water cloud model (WCM). The surface roughness parameters were estimated from the Oh and Baghdadi models. A feed-forward artificial neural network (ANN) method was tested for its potential to translate SAR backscattering and surface roughness input variables to RSM values. The model was trained for three inversion configurations: (i) SAR backscattering from vertical transmit and vertical receive (SAR VV) polarization only; (ii) using SAR VV and the standard deviation of surface heights ($h_{rms}$), and (iii) SAR VV, $h_{rms}$, and optimal surface correlation length ($l_{eff}$). Field-measured volumetric RSM data were used to train and validate the method. The results showed that the ANN soil moisture estimation model performed reasonably well for the estimation of RSM using the single input variable of SAR VV data only. The ANN prediction accuracy was slightly improved when SAR VV and the surface roughness parameters ($h_{rms}$ and $l_{eff}$) were incorporated into the prediction model. Consequently, the ANN’s prediction accuracy with root mean square error (RMSE) = 0.035 cm^3^/cm^3^, mean absolute error (MAE) = 0.026 cm^3^/cm^3^, and *r* = 0.73 was achieved using the third inversion configuration. The result implies the potential of Sentinel-1 SAR data to accurately retrieve RSM content over an agricultural site covered by stubbles. The soil roughness parameters are also potentially an important variable to soil moisture estimation using SAR data although their contribution to the accuracy of RSM prediction is slight in this study. In addition, the result highlights the importance of combining Sentinel-1 SAR and Landsat images based on an ANN approach for improving RSM content estimations over crop residue areas.

## 1. Introduction

Soil water plays an important role in agriculture development and its availability restricts the production of crops throughout the year. In particular, in Ethiopia, where agriculture is highly reliant on rain-fed systems and poor irrigation facilities, the majority of crops are cultivated in summer rainfall, called the Meher season [1]. Residual soil moisture (RSM), which is left in the soil following the harvest of main season cropping, could provide an opportunity to produce additional food and feed crops in the off-season in areas that receive an adequate amount of rainfall. The Upper Blue Nile (UBN) basin in Ethiopia receives annual rainfall >2000 mm [2,3] and after the harvest of main season cropping a certain amount of moisture is left in the soil, which could be used for additional medium or short cycle cropping. Multi-temporal monitoring of moisture in the off-season is required to determine the extent of residual moisture available in the soil. Accordingly, the invention of various techniques and methods to measure and monitor soil moisture from space is essential. In this connection, remote sensing, using both active and passive sensing sensors, has demonstrated a strong potential for estimating the surface soil moisture [4,5,6,7,8]. In the recent past, active microwave remote sensing systems have been preferred by the remote sensing community, primarily due to the sensitivity of synthetic aperture radar (SAR) to surface soil moisture and the availability of SAR data with high spatial and temporal resolution [9,10]. In particular, the Sentinel-1 (S1) satellite mission (composed of S1-A and S1-B constellation) potentially provides SAR data at 20-m nominal spatial resolution at every 3 to 12 days, for different regions [11].

Soil moisture retrieval using SAR signals is strongly overwhelmed by surface roughness and vegetation cover, however, and these parameters affect the behavior of the SAR backscattered signal [12,13]. Subsequently, the effect of these parameters should be removed or minimized to obtain the full sensitivity of SAR data from soil moisture [14]. Different models, such as the statistical Oh [15,16,17] and Dubois [18], the physical (the integral equation model (IEM) [19] and advanced IEM models [20]), and Baghdadi empirical models [21] have been developed to extract soil moisture mainly from bare lands. However, due to the multiple scattering effects of vegetation, these models may not be directly used in vegetation covered areas [22]. In this regard, researchers have developed and applied the semi-empirical water cloud model (WCM) to separate the contribution of vegetation backscatter [23,24,25,26,27] and estimate soil moisture with better accuracy. Based on these models, various soil moisture retrieval models have been developed and tested for multiple SAR satellites operated at the C-band [28,29,30,31,32], X-band [33,34,35], and L- band [36,37,38], and have achieved promising results. For example, Zribi et al. [39] estimated soil moisture in semiarid regions with prediction accuracy of RMSE = 0.06 m^3^/m^3^ using the C-band SAR data and the WCM. He et al. [40] reported a better prediction accuracy of RMSE = 0.033 m^3^/m^3^ in an alpine grassland area through combining the IEM and WCM. Indeed, Tomer et al. [41] found a RMSE ranging from 0.02 to 0.06 m^3^/m^3^ using multi-temporal RADARSAT-2 data. Similarly, the findings of some recent studies [31,42] showed the potential of the newly available Sentinel-1 SAR data to estimate soil moisture. For example, Gao et al. [31] proposed a soil moisture prediction model with retrieval accuracy of RMSE = 0.059 m^3^/m^3^ through combining Sentinel-1 SAR and Sentinel-2 optical data. On addition, Bai et al. [10] reported RMSE = 0.064 m^3^/m^3^ using Sentinel-1 SAR data over the Tibetan Plateau.

Surface roughness in radar applications is expressed by the standard deviation of surface heights ($h_{rms}$) and surface correlation length (l) parameters [43]. The $h_{rms}$ and l represent the vertical and horizontal scale of surface roughness, respectively. Thus, the inversion of soil moisture using SAR data needs the estimation/measurements of both $h_{rms}$ and l. However, most statistical models ignore the effect of l due to the uncertainties in the estimations of l, often resulting in significant inaccuracies in the retrieved soil moisture values [44,45]. To reduce the inaccuracy of soil moisture prediction models, Baghdadi et al. [46,47] calibrated backscattering models to obtain optimum or effective values of parameter l that prevail over the uncertainties related to its ground measurement. After subsequent calibration of the model using different SAR configuration (incidence angles from 23 to 57°, horizontal transmit and horizontal receive (HH), horizontal transmit and vertical receive (HV), and vertical transmit and vertical receive (VV) polarizations) and over different roughness conditions, Baghdadi et al. [48] proposed a model to obtain effective (optimum) l values. Their results revealed that effective l values were a function of $h_{rms}$ values and of radar configuration according to an exponential law. Furthermore, Baghdadi et al. [48] model result was further validated by [49,50] using RADARSAT-1 and X-band SAR data, respectively, and measured datasets for a surface roughness parameter. Accordingly, [49,50] reported the potential of the Baghdadi model to replace the approximation of correlation length measurements and effectively compensate for the inaccuracy of IEM backscattering model. Álvarez-Mozos et al. [49] argued that the Baghdadi model is an important step towards operational radar-based soil moisture estimation.

In addition to the above-mentioned inversion models, the complexity and non-linearity of retrieval problems [51] need the application of more advanced techniques, such as the artificial neural network (ANN). The ANN is a model-free estimator and can be trained to learn the non-linear input-output relationships [52].This model provides an alternative to the classical inversion techniques, which sometimes are restricted by the rigid normality and linearity [53], and has been successfully used for soil moisture estimation in previous studies [54,55]. For example, Satalino et al. [56] retrieve soil moisture from the European Remote Sensing (ERS) SAR data and an ANN approach with an overall accuracy of RMSE of 6%. Similarly, Santi et al. [57] found an RMSE close to 0.023 m^3^/m^3^, using Environmental Satellite (ENVISAT) SAR data and the ANN technique. The potential of ANN modeling for estimating surface soil moisture has been compared to other approaches such as Bayesian and multivariate regression methods [58,59,60]. The result indicated that ANNs are a good substitution in terms of accuracy and stability with respect to the other inversion strategies.

Although a sizeable number of studies have been conducted to estimate soil moisture using SAR data, some topics still need further research work. Most previous research studies that have reported on the subject of soil moisture estimation based on SAR data have mainly focused on bare land or growing croplands. Retrieval of residual soil moisture using SAR data in typical agricultural sites covered by crop residues has not often been reported in the literature [61,62]. Kaojarern et al. [61] proposed a soil moisture retrieval model in post-harvest rice areas using C-band radar imagery in North Thailand, but only irrigation sites were taken into consideration. McNairn et al. [62] analyzed the sensitivity of radar backscatter to post-harvest crop residue in Canada, but their study was not extended to estimate the residual soil moisture values. Nonetheless, the experiments of McNairn et al. [62] demonstrated that crop residue can hold a significant amount of moisture and that residue is not transparent to incident microwaves. Further research work is needed to improve the application of radar sensors to retrieve residual soil moisture content over rainfed agricultural sites covered with crop residues. In addition, different scholars have recommended that the retrieval performance of Sentinel-1 SAR still needs more evaluation work at different sites and for different soil conditions [63].

Landsat datasets were used in the WCM to reproduce the contributions of vegetation water content over the total SAR backscattering signals in crop residue areas. In addition, the inversion of soil moisture from Sentinel-1 SAR observation still requires the measurement or estimation of the two roughness parameters ($h_{rms}$ and $l_{eff}$). In order to overcome the complexity and uncertainity of measuring the rouhness paramaters over an agricultural surface, the well-established and widely used Oh and Baghdadi models were adapted in this study to estimate $h_{rms}$ and $l_{eff}$, respectively.

This study, therefore, aims to: (1) investigated the potential of Sentinel-1 SAR data for residual soil moisture estimation in agricultural sites covered by crop residues; (2) evaluate the contributions of soil roughness parameters ($h_{rms}$ and $l_{eff}$) for improved residual soil moisture monitoring at the scale of the agricultural and expermental plot level; (3) investigate whether the WCM, Oh, and Baghdadi models could be used in agricultural sites covered by crop residues in the UBN basin, Ethiopia; and (4) test the potential of a non-linear ANN technique to translate SAR data, $h_{rms}$, and $l_{eff}$ input data to residual soil moisture values. 

Our paper is organized into five sections. Section 2 presents the materials and methods used in proposed study. Section 3 describes the results of the study. Major findings of the study are discussed in Section 4. Finally, Section 5 addresses the main conclusions.

## 2. Materials and Methods

### 2.1. Site Description

An experimental site with a total area of 400 ha was selected in the Ribb Watershed, located in the Upper Blue Nile (UBN) Basin of Ethiopia. The geographical location of the study site ranges from 11°51′18″ to 11°52′22″ N latitude and 38°11′9″ to 38°12′16″ E longitude (Figure 1). The site has a relatively uniform slope and is dominated by wheat crop residues. The annual climate can be divided into two seasons (i.e., rainy and dry). The rainy season can be split into a short rainy season from February to May and a main rainy season from June to September. The dry season occurs between October and January. The mean annual precipitation and temperature in the study site are about 1295 mm and 20.4 °C, respectively. 

### 2.2. Datasets

#### 2.2.1. Remotely Sensed Images

In this study, remotely sensed images (i.e., the microwave mission of Sentinel-1 A and the optical sensor of Landsat 7 and 8) and in situ based measured data from experimental plots were used. First, open source-based Sentinel-1 SAR image data were acquired from Global Monitoring for Environment and Security (GMES) via the European Space Agency (ESA) website [64] and used for soil moisture estimation. The Sentinel-1 satellite operates a C-band SAR instrument with frequency of 5.405 GHz. The satellite provides SAR data with four different modes, including the main operational Interferometric Wide-Swath (IWS) mode.

Five level-1 products of IWS mode generated as Ground Range, Multi-Look, and Detected (GRD) were acquired from 22 November 2016 to 2 February 2017. Table 1 provides the incidence angle, orbit, and acquisition time for SAR IWS mode. The satellite has an average temporal interval of 12 days in the study area. The GRD product of high-resolution class has a spatial resolution of 20 × 5 m and a pixel spacing of 10 m. Sentinel-1 Team [65] provided the detailed descriptions and characteristics of Sentinel-1 IW swath mode datasets.

The preprocessing of SAR data consists of several steps, including radiometric correction, speckle filtering, and geometric correction. These processes were conducted using the Sentinel Application Platforms (SNAP), open source software provided by European Space Agency (ESA). The calibrations of raw SAR data were undertaken using the radiometric toolbox in SNAP. Radiometric calibration is required to convert SAR pixel values to exact backscattering coefficient of the scene. A 3 × 3 Lee filtering window was employed for the SAR data to reduce the speckles that may degrade the quality of the SAR image. The geometry of the SAR data was corrected using the Range Doppler Terrain Correction Tool in SNAP. Image acquisition in this study was conducted over an incidence angle ranging from 35.7–39.0° (Table 1). However, over large areas and with very different incidence angles, normalization of radar signal is important to correct for variation in backscatter signals due to the variability in the incidence angles.

In addition, optical data of five Landsat images (from both Landsat-7 and Landsat-8 missions) on the same day or within one day after the Sentel-1 SAR data acquisition were acquired from the United States Geological Survey (USGS) website [66] (Table 2). Landsat remotely sensed imageries were used to derive the vegetation water content (*VWC*). In order to estimate the effect of vegetation water content on SAR signals (using WCM), ancillary data were extracted from optical satellites. 

Following the failure of the Scan Line Corrector (SLC) of Landsat 7 Enhanced Thematic Mapper plus (ETM+) in 2003, Landsat 7 ETM+ images have wedge-shaped gaps, resulting in data loss. Thus, the scan line error of Landsat 7 ETM+ in this study was handled using the “Fill nodata” tool in QGIS 3.6. The reflectance values of near-infrared (*NIR*) and short-wave infrared (*SWIR*) bands were used to calculate the normalized difference water index (*NDWI*). Then, the *NDWI* values for each sample point were derived and combined with field measurements to establish the relationship between vegetation water content (*VWC*) and *NDWI*.

#### 2.2.2. Experimental Ground Measurements

Multi-temporal ground measurements, such as surface soil moisture and crop residue water content, were collected simultaneously with the acquisition of Sentinel-1 SAR data from 14 sampling plots. Seventy in situ soil moisture measurements were obtained during the five field visits from 22 November 2016 to 2 February 2017. Each sampling plot had an area of 900 m squared and contained wheat residue. Positional coordinates of the sampling plots were collected using a Global Positioning System (GPS) receiver. The sampling plots were selected based on plot homogeneity and uniform slope while maintaining a reasonable accessibility. Considering moisture variability at the plot scale, three surface soil moisture measurements using an ECH_2_O EC-5 sensor were made in each of the measurement plots at a depth of 5 cm. These measurements were averaged to obtain the plot average soil moisture. The ECH_2_O EC-5 is a Frequency Domain Reflectometry (FDR) sensor, which provides volumetric (cm^3^/cm^3^) soil water content measurements. The comparison of ECH_2_O EC-5 volumetric measurement with the gravimetric method resulted in strong linear relationships with *r* = 0.94 and a RMSE of ±0.035 (cm^3^/cm^3^). The significance of crop residue to radar signals is highly dependent on the amount of water it contains [62,67]. The authors concluded that residue cover will obstruct the use of radar sensors for soil moisture mapping. Thus, water contents of wheat crop residues were measured at 38 sampling points. The above ground biomass within an area of 0.5 × 0.5 m was harvested and the weights of residue before and after being place in a drying oven were used to calculate the residue water content of each sampling points. 

### 2.3. Methods

#### 2.3.1. Parameterization of Crop Residue Effect

The vegetation water content (*VWC*) is one of the most significant time and space varying parameters of vegetation that reduces the sensitivity of radar measurements to soil moisture [68,69]. In our study, SAR backscatter values were acquired from agricultural plots with wheat crop residues, and it was expected that residue water content would affect the backscattering characteristics of soil [62,67]. A number of precise models have applied to simulate the effect of vegetation in a variety of situtations over different vegetation type and soil conditions. The widely used semi-emperical water cloud model (WCM) [70] was applied in this study to seperate the crop residue water contribution from the radar signal. The WCM assumes that vegetation is a source of homogeneous scattering. The total radar backscattering coefficient ($\sigma^{{}^{\circ}}$) from a canopy can be expressed as the incoherent sum of contribution due to volume scattering ($\sigma_{veg}^{{}^{\circ}}$) from the vegetation canopy itself, double-bounce scattering components between the vegetation and the underlying soil surface ($\sigma_{veg+soil}^{{}^{\circ}}$), and direct soil backscattering ($\tau^{2}\sigma_{soil}^{{}^{\circ}}$) attenuated by vegetation, where *τ*^2^ is the two-way attenuation of vegetation layer. Thus, for a given incidence angle (θ), the WCM can be written as follows (in units of dB).
(1)$$\sigma^{{}^{\circ}}=\sigma_{veg}^{{}^{\circ}}+\sigma_{veg+soil}^{{}^{\circ}}+\tau^{2}\sigma_{soil}^{{}^{\circ}}$$

In addition, the model assumes that the effect of the interactions between vegetation and soil are insignificant and could be neglected in the WCM [71]. Therefore, the WCM can be reformulated as follows:(2)$$\sigma^{{}^{\circ}}=\sigma_{veg}^{{}^{\circ}}+\tau^{2}\sigma_{soil}^{{}^{\circ}}\;\tau^{2}=\exp(-2Bm_{vwc}\sec\theta )\;\sigma_{veg}^{{}^{\circ}}=Am_{vwc}\cos\theta (1-\tau^{2})$$
where the total backscattering coefficient ($\sigma^{{}^{\circ}}$) was observed from Sentinel-1 SAR mission and $m_{vwc}$ is the field-measured *VWC* (kg/m^2^). Accurate estimation of A and B requires prior information about the water content of the vegetation. In addition, an experimental dataset generated from a theoretical model is required to determine A and B parameters. Unfortunately, the simulation of this dataset from the theoretical model requires in situ surface roughness parameters, which we did not have in this study due to the complexity of collecting the datasets relating to the agricultural surface and resource limitations. The surface roughness parameters in this study were estimated from the well-established and widely used models. Accordingly, [22] calculated the correction values of A and B under different underlying surfaces; these are provided in Table 3. These parameters were used by Huang et al. [25] to retrieve soil moisture using Sentienel-1 over sparse vegetation coverage.

In addition, the *VWC* of the study area was estimated by combining field-measured vegetation water content and normalized difference water index (*NDWI*). Compared to other vegetation indices, the *NDWI*-based method for *VWC* estimation has been found to be superior based upon a quantitative analysis of bias and standard error [72,73]. Thus, the relationship between *VWC* and *NDWI* was developed using the least-square fitting approach as follows [23,72]:(3)$$VWC=aNDWI^{2}+bNDWI$$
where *a* and *b* are model empirical parameters. *NDWI*, which was formulated by [74], is expressed as: (4)$$NDWI=\frac{NIR-SWIR}{NIR+SWIR}$$
where *NIR* and *SWIR* are the reflectance or radiance corresponding to the near infrared and short-wave infrared wavelength channels, respectively. The *NDWI* value varies between −1 to +1, depending on the water content of the vegetation. Subsequently, the bare soil backscattering coefficients ($\sigma_{soil}^{{}^{\circ}})$ can be computed using Equation (5). According to the underlying vegetation type in this study, the parameters for A and B were selected from Table 3, namely A = 0.0018, B = 0.138.
(5)$$\sigma_{soil}^{{}^{\circ}}=\frac{\sigma^{{}^{\circ}}-0.0018\times m_{vwc}\cos\theta [1-\exp(-0.276\times m_{vwc}\sec\theta )]}{\exp(-0.276\times m_{vwc}\sec\theta )}$$

#### 2.3.2. Estimation of Soil Roughness Parameters 

The $\sigma_{soil}^{{}^{\circ}}$ was estimated from the WCM to eliminate the contributions of crop residue and it contains the backscattering of soil moisture and surface roughness. Thus, incorporating the effect of surface roughness is vital to monitor surface soil moisture with good accuracy. The surface roughness is expressed by the standard deviation of surface heights ($h_{rms}$) and surface correlation length (l) parameters [43]. The Oh [[15]−[17]] semi-empirical backscattering model is a suitable model to estimate $h_{rms}$. The model relates the co-polarized ratio and the cross-polarized ratio to incident angle (θ), wavenumber (k), $h_{rms}$, and volumetric soil moisture ($m_{v}$). In this study, the Oh model [17] with cross-polarized ratio was used to estimate the $h_{rms}$. The Oh model was used by [75] to estimate the surface roughness parameter. It is formulated as follows:(6)$$q=\frac{\sigma_{soil\_VH}^{{}^{\circ}}}{\sigma_{soil\_VV}^{{}^{\circ}}}=0.095{\left(0.13+sin1.5\theta \right)}^{1.4}\left(1-e^{-1.3\left(k.h_{rms}\right)0.9}\right)$$

The algorithm is optimized for bare soils with 0.1 $\leq k.h_{rms}\leq 2.5,$ 9% ≤ soil moisture (mv)≤ 31% and $10^{{}^{\circ}}\leq \theta \leq 70^{{}^{\circ}}$. The direct inversion model for $h_{rms}$ is:(7)$$h_{rms}=\frac{{\left\{\frac{-1}{1.3}ln\left[1-\frac{q}{0.095{\left(0.13+sin1.5\theta \right)}^{1.4}}\right]\right\}}^{1.111}}{k}$$

The surface correlation length parameter is estimated using a model proposed by Baghdadi et al. [48], which is a function of $h_{rms}$:(8)$$l_{eff}(h_{rms},\;\theta ,\sigma_{soil\_VV}^{{}^{\circ}})=\alpha .h_{rms}{}^{\beta }$$
where $l_{eff}$ refers to effective l, θ the incidence angle, and α and β are coefficients that depend on θ and $\sigma_{soil\_VV}^{{}^{\circ}}$ and can be calculated as follows:(9)$$\alpha_{\sigma_{soil\_VV}^{{}^{\circ}}}=\delta {\left(\sin\theta \right)}^{\mu }\;\beta_{\sigma_{soil\_VV}^{{}^{\circ}}}=\eta \theta +\xi$$
where δ, μ, η, and ξ are calibration coefficients. δ and ξ are dependent on the polarization, while μ and η were found to be independent:$$\delta_{VV}=3.289,\;\xi_{VV}=1.551,\;\mu =-1.744,\;\mathrm{and}\;\eta =-0.0025.$$

From the five Sentinel-1 SAR data acquisition periods, SAR data with both VV and VH polarization were acquired for descending orbit path on 22 November 2016, 16 December 2016, and 2 February 2017. Consequently, $h_{rms}$ and $l_{eff}$ were calculated for this period. However, given the SAR data were acquired during the offseason, we assumed that temporal changes in soil surface roughness caused by agricultural activities, such as tillage, and rainfall events are minimal. Thus, the average $h_{rms}$ and $l_{eff}$ values were taken for all the analysis of the study periods, except for 2 February 2017 when the study site received a shower of rain during the final week of January 2017. The roughness parameters during this period were estimated separately.

#### 2.3.3. Artificial Neural Network (ANN)

Three input variables, namely, bare soil backscattering coefficients from VV polarization ($\sigma_{soil\_vv}^{{}^{\circ}}$) and surface roughness parameters ($h_{rms}$ and $l_{eff}$) were used to train the prediction model. The Oh and Baghdadi models have provided an opportunity to estimate $h_{rms}$ and $l_{eff}$ at spatial level (i.e., corresponding each pixel of the SAR image) in our study site, which could be used to produce soil moisture maps for the trained model.

The SAR backscattering and soil roughness parameter values corresponding to each sampling plot were extracted and used as an input parameter to the ANN. In addition, to evaluate the relative performance of the ANN approach, a linear regression model (LRM) was also trained. The datasets were separated into two parts, i.e., training and validation datasets. Then, for the experimental plots, 70% of the sampling points were used as training data sets; the remaining 30% was used for validation. Both the ANN and LRM were developed using the same training datasets. Three inversion configurations based on SAR backscattering and soil roughness parameters were defined: (1) $\sigma_{soil\_vv}$, (2) $\sigma_{soil\_vv}$ and $h_{rms}$, and (3) $\sigma_{soil\_vv}$, $h_{rms}$ and $l_{eff}$. Each method was trained for the three inversion configurations. The statistical packages included in R software were used in this study. The schematic diagram presented in Figure 2 shows the soil moisture retrieval algorithms used in this study.

The ANN can imitate human learning capabilities and develop multivariate nonlinear relationships, and is thus widely applied for estimating land surface parameters from remote sensing data [76]. An ANN analysis is built from a number of hidden neurons nodes that work side-by-side to convert data from input layers to output layers. Each ANN has a two-phase process: the training and validation phases. In the training phase, each neuron is trained using the training sample dataset as an input variable pattern to produce an output pattern. In the validation phase, when an input pattern is fed to the model, the ANN will produce its associated output values [77]. In this study, a feed-forward multilayer perceptron (MLP) neural network model was applied to transform a set of input variables into a set of output variables. Figure 3 shows the fundamental ANN structure consisting of input layers, a hidden layer, and an output layer.

In a typical neural network model (Figure 3), a neuron contains a weighted sum of the input variables ($x_{1},x_{2},\ldots ,x_{m}$) and transforms this sum using a non-linear function to provide the final output as follows:(10)$$y_{k}=\varphi \left(u_{k}+b_{k}\right)$$
where $x_{1},x_{2},\ldots ,x_{m}$ are the inputs signals (variables); $w_{k_{1}},w_{k_{2}},\ldots ,w_{k_{m}}$ are the respective weights of neuron *k*; $u_{k}$ is the linear combination output due to the input variable; $b_{k}$ is the bias; φ(.) is the activation function; and $y_{k}$ is the output.

The SAR backscattering coefficient ($\sigma_{soil\_vv}$), and the soil roughness parameters ($h_{rms}$ and $l_{eff})$ are the input variables; the corresponding volumetric soil moisture is the output variable. Thus, the ANN model was trained for three different inversion configurations using the “neural net” package in R software. All the configurations lead to a one-dimensional output layer that contains volumetric surface soil moisture. For the optimization of the ANN parameters (hidden layer and hidden nodes), many experiments were conducted.

Based on this optimization process, the MLP architecture was determined to have a single hidden layer neural network with three hidden nodes (for the first inversion configuration with a single input variable, $\sigma_{soil\_vv}$), six hidden nodes (for the second configuration with two input variables, $\sigma_{soil\_vv}$ and $h_{rms}$), and 10 hidden nodes (Figure 4) (for the third configurations with three input variables, $\sigma_{soil\_vv}$, $h_{rms}$ and $l_{eff}$) to predict residual soil moisture.

The performance of the prediction models was investigated using the root mean square error (RMSE), mean absolute error (MAE), the bias, and the correlation coefficient (*r*) based on the R statistical packages.

## 3. Results

### 3.1. Crop Residue Water Content

Information about the crop residue water content is an important parameter of the WCM to reduce the effect of crop residue on soil backscattering coefficients of SAR data. In this case, the *NDWI* was selected as the predicting index to generate the *VWC* of the entire study site based on the relationship established between Landsat surface reflectance data and ground-based *VWC* measurements. Then, the least-square method (Equation (3)) was used to calculate coefficients (a = 10.33 and b = −0.40) of the fitting model and resulted in the correlation coefficient of *r* = 0.87. The *VWC* map estimated for the entire study site for the five temporal periods using the proposed model and the Landsat data is presented in Figure 5. Observing the spatial and temporal distributions of the *VWC* over the study period, generally, the amount of *VWC* reduced from November 2016 to February 2017. The amount of *VWC* in our study site could, however, depend on the geometry, the height and density of wheat crop residues, and the proximity of particular plots to permanent plantations. A permanent plantation may control the evaporation process of the nearby plots through its shading effects. The *VWC* value from the study site ranges from 0.32 to 0.69 kg/m^2^.

### 3.2. The Relation Between Radar Backscattering Coefficient of Bare Soil and Soil Moisture.

The relatively high spatial resolution Sentinel-1 SAR data provided an opportunity to analyze soil moisture at the agricultural plot scale. As an initial step, a sensitivity analysis of theSentinel-1 SAR backscatter coefficient ($\sigma^{{}^{\circ}}$) and in-situ measured residual soil moisture was conducted to verify the potential of Sentinel-1 SAR data to retrieve soil moisture in the wheat stubble agricultural fields (Figure 6a). Over the periods of this study, the soil moisture varied between 0.07 and 0.24 (cm^3^/cm^3^) and while the radar backscatter signals ranged from −16.53 to −10.58 dB (Figure 6a). As shown in Figure 6a, Sentinel-1 $\sigma_{vv}{}^{{}^{\circ}}$ data have shown a positive correlation (*r* = 0.38) with measured soil moisture. Overall, our result is consistent with previous findings [9,79]. The low correlation of the linear model could be attributed to the effect of residue water content and soil surface roughness [80,81], which attenuate and scatter the electromagnetic radiation. Indeed, incorporating the effect of vegetation water content and surface roughness parameters is an important practice proven in previous studies for reliable soil moisture retrieval using SAR data. 

As discussed in Section 2.3.1, the effect of crop residue water content on the Sentinel-1 SAR backscatter coefficient is introduced into the semi-empirical model of the WCM. Then, the backscattering of bare soil for the study site was estimated using Equation (5). The correlations of SAR backscatter coefficient ($\sigma^{{}^{\circ}}$) and soil backscatter ($\sigma_{soil}^{{}^{\circ}}$) to field-measured residual soil moisture were also compared to observe the perturbing effect of vegetation to SAR backscattering signals (Figure 6a,b). Thus, reducing the effect of crop residue water content using WCM, which impedes the backscatter signals of the underling soil surface, improved the correlation coefficient between SAR backscatter and measured soil moisture to *r* = 0.54 (Figure 6b). Figure 6, in general, reveals the importance of reducing the perturbing effects of vegetation and the reliability of the WCM model to reduce these effects in the retrieval of soil moisture over stubble agricultural sites. Nonetheless, bare soil backscattering coefficients are composed of the scattering from surface roughness and soil moisture. Accurate retrieval of soil moisture using SAR data is highly dependent on the ability to reduce the effects of the backscatter coefficients of surface roughness from bare soil backscatter. In this study, the Oh and Baghdadi models were used to estimate the $h_{rms}$ and $l_{eff}$, respectively.

### 3.3. Estimating Surface Roughness Parameters

The soil roughness properties of a natural surface are described by $h_{rms}$ and l. In the absence of field-measured surface roughness data, the Oh model (Equation (6)) is the appropriate method to estimate the surface roughness parameter. The result indicates that the $h_{rms}$ of the surface in our study site ranged from 1.30 to 2.92 cm (Figure 7). 

As presented in Figure 7, $h_{rms}$ was calculated for 22 November 2016 (Figure 7a), 16 December 2016 (Figure 7b), and 2 February 2017 (Figure 7c), using SAR data acquired both for VV and VH polarizations. Since the SAR data were acquired during the offseason in the study area, changes in the soil roughness due to agricultural activities and rainfall very minimal, except for the rain shower observed during the final week of January 2017. Consequently, $h_{rms}$ during 2 February 2017 (Figure 7c) was high relative to the other dates. As a result, the average $h_{rms}$ (Figure 7d) calculated from Figure 7a,b was used in the analysis for the other dates, with the exception of 2 February 2017, which used $h_{rms}$ calculated from the same date. The varying distributions of $h_{rms}$ values shown over the study area could be attributed to the difference in the plowing practices among farms and the direct contact of the soil surface during rainy events. The result presented in Figure 7 may show the reliability of the Oh model in estimating the $h_{rms}$ in our study site. 

However, the correlation length (*l*) is a difficult parameter to determine and was not estimated from SAR data using the Oh model due to the insensitivity of the cross-polarization ratio on correlation length. The model developed by [48] was used to calculate the effective correlation length ($l_{eff}$) and the results are provided in Figure 8. The $h_{rms}$ calculated from the Oh model in Figure 7c,d was used to estimate the effective correlation length based on the model proposed by Baghdadi [48]. The results indicate that the correlation length of the study area ranged from 9.74 to 17.2 cm and had a similar spatio-temporal pattern to that of $h_{rms}$. Moreover, the correlation length increased as the $h_{rms}$ of the surface increased.

### 3.4. Soil Moisture Estimation

The relationship between measured and predicted residual soil moisture was analyzed using MAE, RMSE, bias, and the coefficient of correlation (*r*) (Table 4). In addition, the scatter plots between measured and predicted soil moisture for both the ANN and LRM trained with the three input variables $\sigma_{soil\_vv}$ (SAR backsacttering of bare soil from the VV polarization), $h_{rms}$ (the standard deviation of surface heights) and $l_{eff}$ (effective correlation length) are shown in Figure 9. In this study, the point measurements at the agricultural plots are assumed to represent the average residual soil moisture in the area corresponding to SAR data. In general, results from Table 4 indicate that the prediction models developed based on ANN and LRM methods produced a good agreement with the measured soil moisture data in terms of MAE, RMSE, bias and *r*. Generally, the soil moisture retrieval accuracy increases with an increase in input variables, although the improvement made in this aspect is very slight. However, both the LRM and ANN models showed a satisfactory performance in predicting volumetric soil moisture using $\sigma_{soil\_vv}$ as a single input variable, with the highest correlation (*r* = 0.60) generated by the LRM method. In this case, both the ANN and LRM resulted in RMSE = 0.040 cm^3^/cm^3^ and MAE = 0.030 cm^3^/cm^3^.

Although the addition of surface roughness parameters ($h_{rms}$ and $l_{eff}$) to the prediction model does not show the required improvements in terms of MAE and RMSE, the improvements in the bias and correlation coefficient are encouraging. For example, the bias of −0.034 cm^3^/cm^3^ and −0.032 cm^3^/cm^3^ observed from the first model with SAR VV alone for the LRM and ANN, respectively, was improved to −0.014 cm^3^/cm^3^ and −0.024 cm^3^/cm^3^.

The correlation coefficient was also enhanced to *r* = 0.70 and *r* = 0.73 for the LRM and ANN, respectively. However, with regard to the bias for the third configuration of the input variables, the LRM method relatively overestimates the predicted soil moisture, while the ANN method underestimates it. This can be observed in the soil moisture maps given in Figure 10 and Figure 11. In addition, Figure 9 depicts that both the LRM and ANN models underestimate measured residual soil moisture values greater than 0.20 cm^3^/cm^3^. 

The ANN and LRM prediction models trained with the three input variables $\sigma_{soil\_vv}$, $h_{rms}$, and $l_{eff}$ were applied to pixel-wise input data. Thus, the soil moisture maps of the study site for each prediction model and temporal data were generated to demonstrate the spatio-temporal variability of estimated soil moisture at various dates (Figure 10 and Figure 11). The attributes of the pixels of these maps show the predicted soil moisture. The soil moisture predicted in the study area ranged from 0.05 to 0.36 cm^3^/cm^3^ (Figure 10 and Figure 11). However, regarding the spatio-temporal patterns of the estimated soil moisture in both Figure 10 and Figure 11, the values of the residual soil moisture for almost all farm lands ranged between 0.05 to 0.22 cm^3^/cm^3^. Only very few pixels had insignificant soil moisture values of >0.3 cm^3^/cm^3^, and these values would not show the real behavior of the proposed prediction models. Some of the soil moisture variation observed between the two models (Figure 10 and Figure 11) could be explained by the overestimation of LRM and underestimation of ANN models (Table 4).In general, the temporal patterns of soil moisture assert that soil moisture values were reduced from 22 November to 23 December 2016 and followed the meteorological conditions of the study site. However, the study area regained higher soil moisture values on 2 February 2017 (Figure 10e and Figure 11e) due to rainfall on the previous day, which resulted in the increase of soil moisture in most of the agricultural plots in the study site.

## 4. Discussion

Surface soil moisture is sensitive to radar backscattering and can be derived from SAR data using different methods [58]. However, radar backscattering is also sensitive to other time- and space-varying parameters such as vegetation and soil roughness, in addition to soil moisture [14,33]. In this study, we propose residual soil moisture retrieval algorithms for wheat stubble agricultural sites using Sentinel-1 SAR and Landsat data based on ANN methods. As an initial step, the linear relationship between measured volumetric soil moisture and Sentinel-1 total radar backscatter ($\sigma_{vv}^{o}$) was determined. The results showed the potential of Sentinel-1 SAR data for soil moisture estimation in wheat stubble agricultural fields, although it produced a low *r* value (i.e., *r* = 0.38; Figure 6a). Indeed, the low *r* value might be attributed to the effect of crop residue water content and soil roughness parameters on soil moisture backscattering coefficients. Thus, different scholars [23,60,82] have argued that considering or reducing the effect of vegetation and soil roughness variables can further improve the accuracy of SAR-based soil moisture estimation using non-linear regression models such as the artificial neural network (ANN). In this study, the effect of crop residue water content was addressed through the WCM and then the bare soil backscatter coefficients were estimated. Reducing the effect of crop residue water content resulted in an improved correlation between SAR backscattering coefficient and residual soil moisture with *r* = 0.54. To further improve the soil moisture prediction accuracy, the Oh and Baghdadi models were also adopted to estimate the soil roughness parameters (i.e., $h_{rms}$ and $l_{eff}$) of the study site. The volumetric soil moisture prediction models based on ANN and LRM were trained using the different configurations of $\sigma_{soil\_vv}$, $h_{rms}$, and $l_{eff}$, input variables.

The ANN model showed satisfactory performance when it was trained using a single input variable, $\sigma_{soil\_vv}$, which ignores the effect of soil roughness parameters. It resulted in an RMSE as low as 0.040 cm^3^/cm^3^, MAE = 0.030 cm^3^/cm^3^, bias = −0.032, and *r* = 0.57 (Table 4). This could be due to the removal of the effect of crop residue water content on soil backscattering coefficients. With the same method, Ahmad et al. [82] also produced a satisfactory performance for soil moisture estimation using single polarized SAR (HH) data and the ANN inversion technique. Although this study revealed the importance of soil roughness parameters to enhance the prediction accuracy of soil moisture retrieval models, the contribution made by surface roughness in terms of improving the MAE and RMSE was very slight. This is likely associated with the acquisitions of SAR data outside the growing season, where a change in soil surface roughness caused by rainfall and farming practice is minimal. In fact, the improvement due to using surface roughness parameters on the bias and correlation coefficient is encouraging. Overall, the result indicates the potential of Sentinel-1 SAR data and the ANN method to translate the input variables into volumetric soil moisture, and the feasibility of the Oh and Baghdadi models to estimate the soil roughness parameters for our study site. Similarly, Alexakis et al. [42] and Meng et al. [23] successfully estimated surface soil moisture using Sentinel-1 SAR data and the ANN method, after correcting for the effect of surface roughness and vegetation water content on SAR backscattering coefficients. Similarly, Brogioni et al. [83] obtained an improved ANN-based soil moisture prediction using the C-band backscattering and ancillary soil surface roughness information.

Our result is comparable to previous studies [31,41,44,56,57] found in the literature, and the accuracy reported in this study is acceptable and within the range of previous findings. Indeed, the soil moisture variation observed between the predicted and measured soil moisture shown in Figure 9 could be due to the simplified from of sigma used to represent the intensity of SAR, the uncertainty of the models used, and the varying soil texture properties of the experimental site. Variation in soil texture properties may lead to varying soil water holding capacity and, in turn, affect SAR signal sensitivity. Based on our analysis of the sample plots, the study area can be classified into four major soil texture classes: clay, heavy clay, loam, and clay loam. In addition, the underestimations of both LRM and ANN models over soil moisture values greater than 0.20 cm^3^/cm^3^ could be related to the limited training datasets within this range of soil moisture variation. In general, the findings of this study demonstrated the importance of integrating Sentinel-1 SAR and Landsat data and the surface roughness parameters for the finest prediction of surface soil moisture. The LRM was analyzed to further show the performance of ANN in soil moisture prediction (Figure 9 and Table 4). In general, the ANN showed a slight improvement over the LRM model in terms of accuracy. However, the proposed ANN method and Sentinel-1 SAR data is a reliable approach, and could achieve acceptable performance for high-resolution soil moisture estimation, which could be used for agricultural applications, such as soil moisture monitoring over farmlands.

## 5. Conclusions

In this study, we propose a residual soil moisture prediction model for agricultural fields covered by wheat stubble using Sentinel-1 SAR and Landsat data based on the ANN method. A combination of a semi-empirical backscattering model of soil and vegetation, and empirical relationships derived from Sentinel-1 SAR data and soil roughness parameters, were used to estimate the residual soil moisture. The approach essentially consists of four major steps: (i) estimation of vegetation water content through combining *NDWI* and field measured vegetation water content, (ii) estimation of bare soil backscattering coefficients using the water cloud model (WCM) and Sentinel-SAR data, (iii) calculate the soil roughness parameters ($h_{rms}$ and $l_{eff}$) based on Oh and Baghdadi models, and (iv) inversion of soil moisture values using the artificial neural network (ANN) method.

An optimal ANN model that consists of three input variables (i.e., V, $h_{rms}$, and $l_{eff}$) was developed. The prediction results using field-measured soil moisture revealed that the proposed prediction model achieves reasonable soil moisture estimation accuracy (e.g., RMSE = 0.035 cm^3^/cm^3^). The results also indicated the potential of Sentinel-1 SAR data and the ANN-based prediction model for soil moisture retrieval. Based on our findings, we can conclude that crop residue water content is an important factor for accurate estimation of residual soil moisture in harvested agricultural plots. In addition, incorporating the effect of soil roughness parameters was verified to be important for SAR-based soil moisture prediction, although their contribution to soil moisture prediction accuracy was slight for the study site. In addition, the findings of the study confirmed that the combination of Sentinel-1 SAR and Landsat sensor products as input datasets for the ANN model made a significant contribution to improved soil moisture content estimation. Our results further confirmed that the Oh and Baghdadi models are an important approach to estimating soil roughness parameters when few or no field-measured soil roughness values exist. Although acceptable soil moisture prediction performance was achieved, the study has the following limitations that need to be considered in future studies: (i) the limited number of sample plots observed and (ii) the spatial scale mismatch between ground-observed points and satellite footprints/pixels, although an attempt was made to reduce this error to some extent by averaging multipoint measurements. These limitations could be partly addressed through training the prediction model over a large number of sample plots with dense ground observations. Future research should focus on validating the performance of the proposed soil moisture prediction and the soil roughness estimation models under different climate and land use and land cover conditions using a large number of datasets.

## Figures and Tables

**Figure 1 sensors-20-03282-f001:**
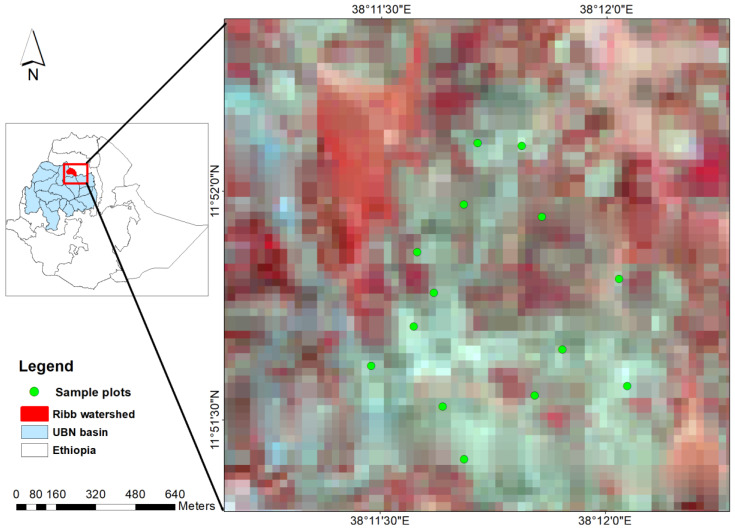
Location map of the study area (background with Land sat satellite imagery) located in the Ribb Watershed of the Upper Blue Nile (UBN) Basin, Ethiopia.

**Figure 2 sensors-20-03282-f002:**
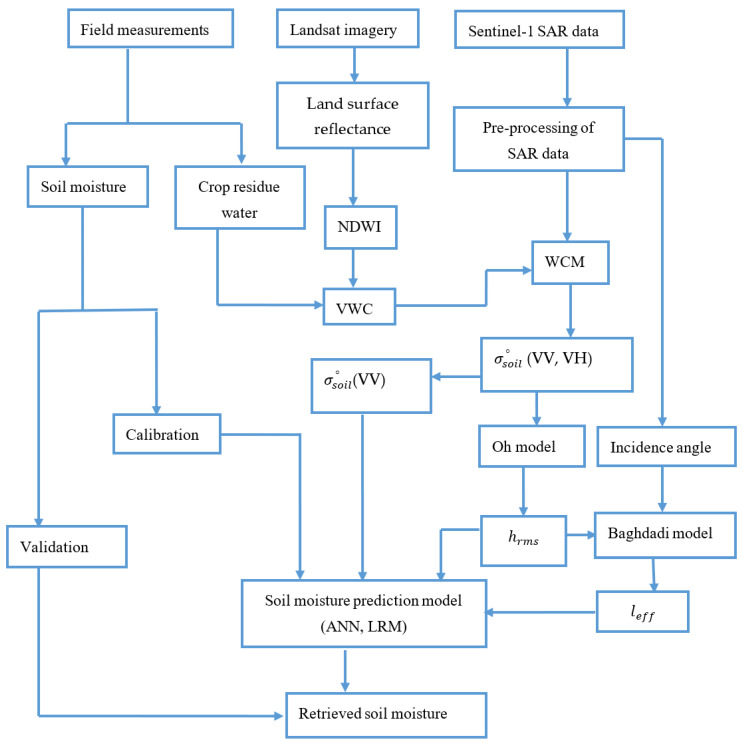
Schematic diagram shows the flow of soil moisture estimation. *NDWI*-normalized difference wetness index, *VWC*-vegetation water content, WCM- water cloud model, VV- vertical transmit and vertical receive polarization, HV- horizontal transmit and vertical receive polarizations, ANN-artificial neural network, and LRM-linear regression model.

**Figure 3 sensors-20-03282-f003:**
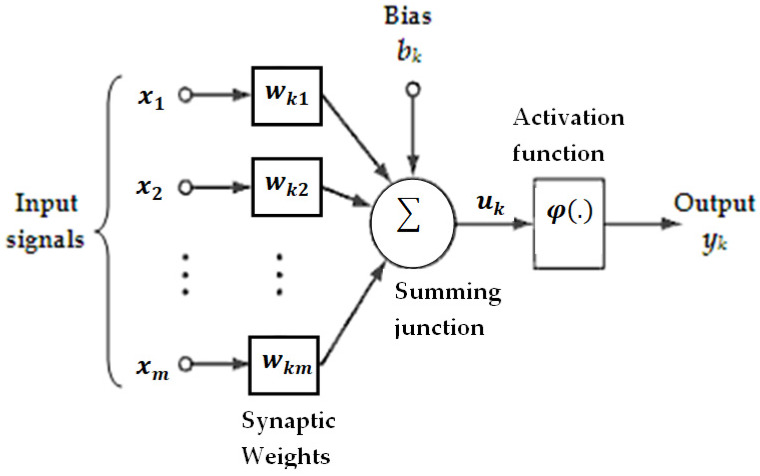
An artificial neuron model structure [78].

**Figure 4 sensors-20-03282-f004:**
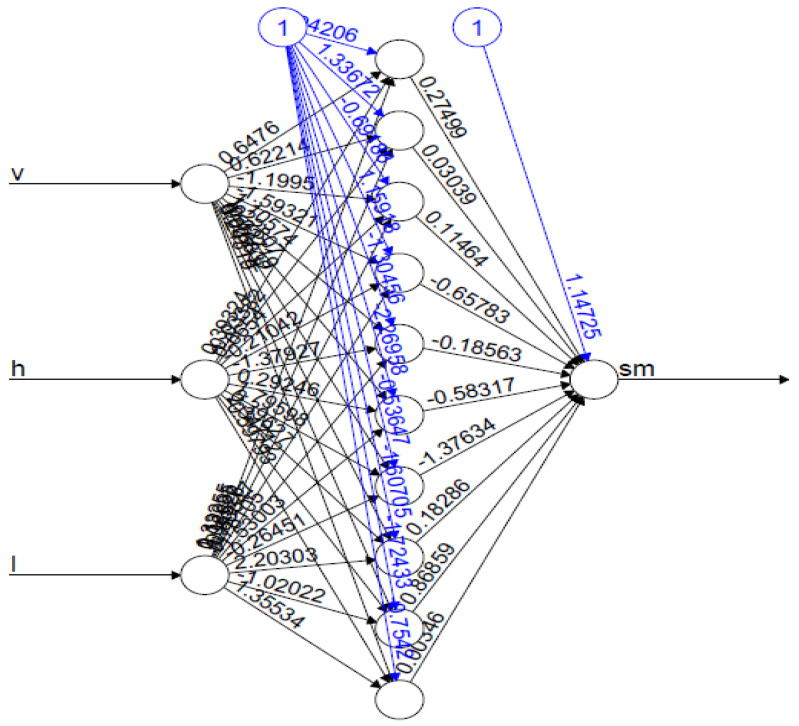
Artificial neural network (ANN) architecture used for soil moisture estimation using v-$\sigma_{soil\_vv}$, $h-h_{rms}$ and $l-l_{eff}$ as input variables and soil moisture (sm) as the output variable. The structure has one hidden layer and 10 hidden neurons.

**Figure 5 sensors-20-03282-f005:**
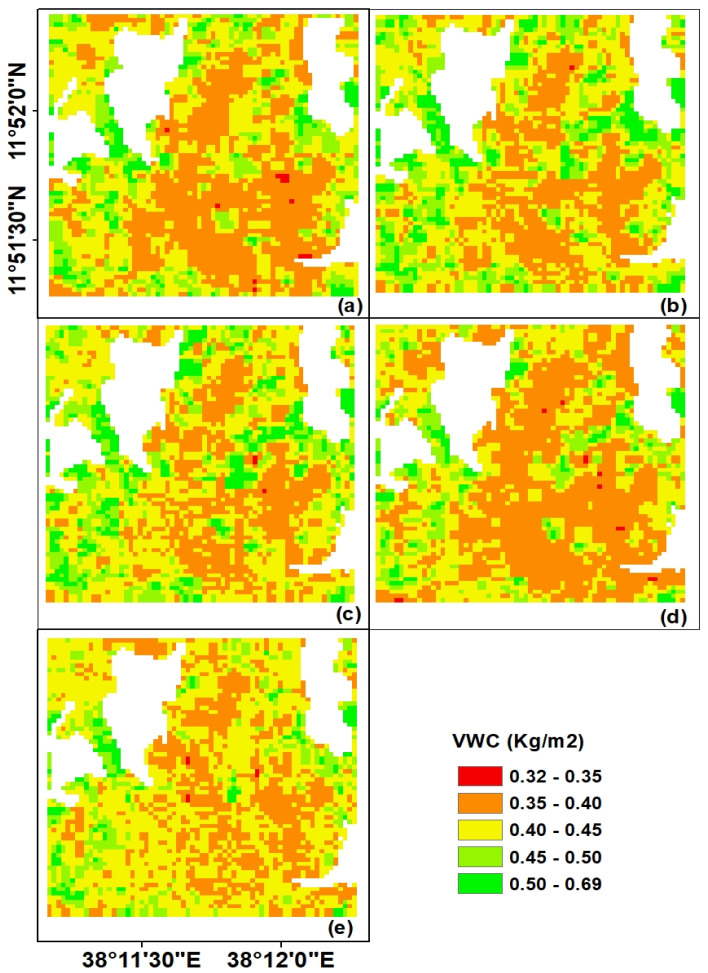
Vegetation water content (*VWC*) maps of the study site derived based on Landsat satellite data: (**a**) 22 November 2016, (**b**) 29 November 2016, (**c**) 16 December 2016, (**d**) 23 December 2016, and (**d**) 2 February 2017.

**Figure 6 sensors-20-03282-f006:**
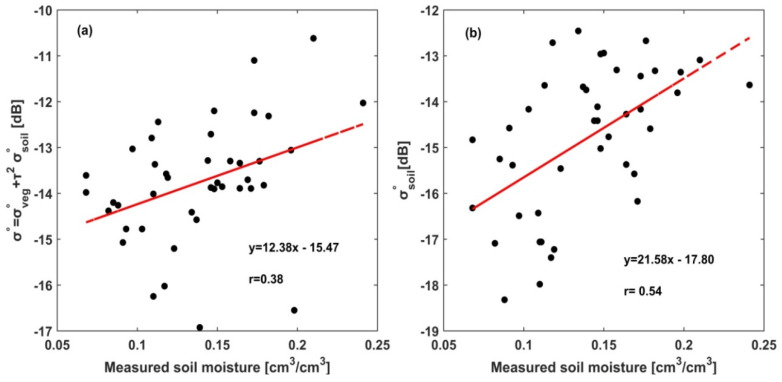
The correlation between (**a**) field-measured soil moisture (at 5 cm depths) and Sentinel-1 radar backscatter ($\sigma^{{}^{\circ}}$) and (**b**) the correlation between field-measured soil moisture and soil backscatter signals ($\sigma_{soil}^{{}^{\circ}}$). The red lines show the linear regression.

**Figure 7 sensors-20-03282-f007:**
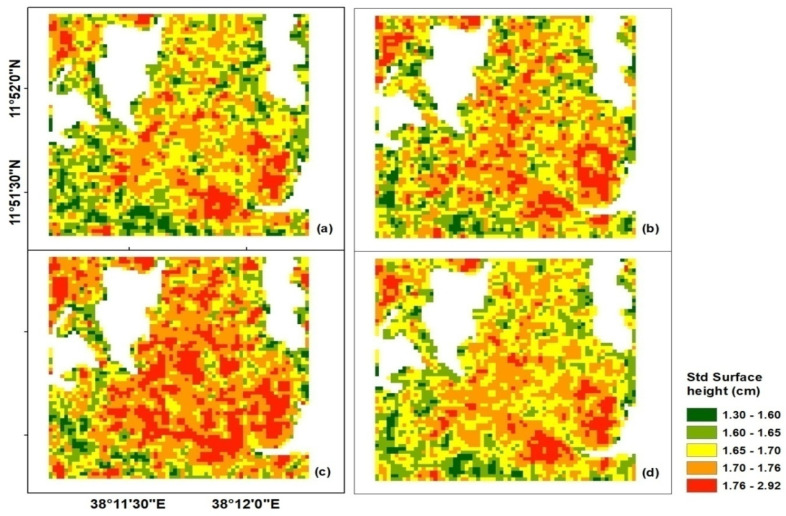
The standard deviation of surface heights ($h_{rms})$: (**a**) 22 November 2016, (**b**) 16 December 2016, (**c**) 2 February 2017, and (**d**) the average $h_{rms}$ calculated from (**a**) and (**b**).

**Figure 8 sensors-20-03282-f008:**
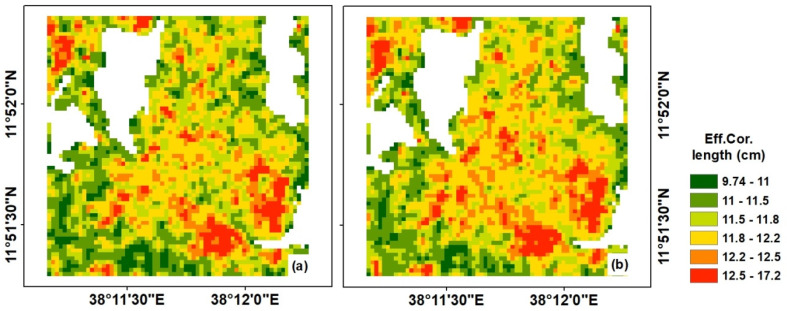
Effective correlation length ( $l_{eff}$) of the surface: (**a**) based on the average $h_{rms}$ and (**b**) based on February $h_{rms}$ values.

**Figure 9 sensors-20-03282-f009:**
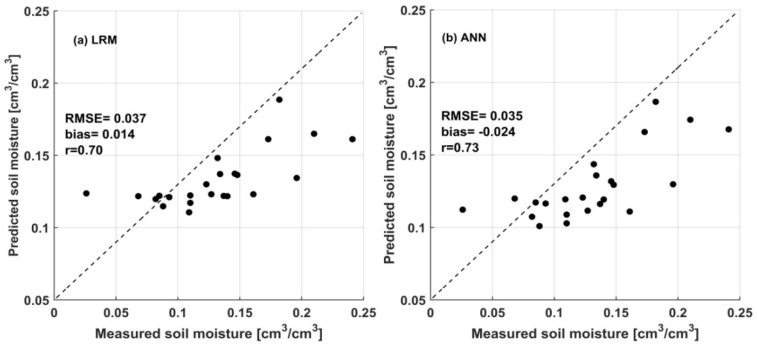
Scatter plots showing the comparison between measured and predicted soil moisture based on the prediction model developed using $\sigma_{soil\_vv}$, $h_{rms}$, and $l_{eff}$ input variables for (**a**) LRM and (**b**) ANN.

**Figure 10 sensors-20-03282-f010:**
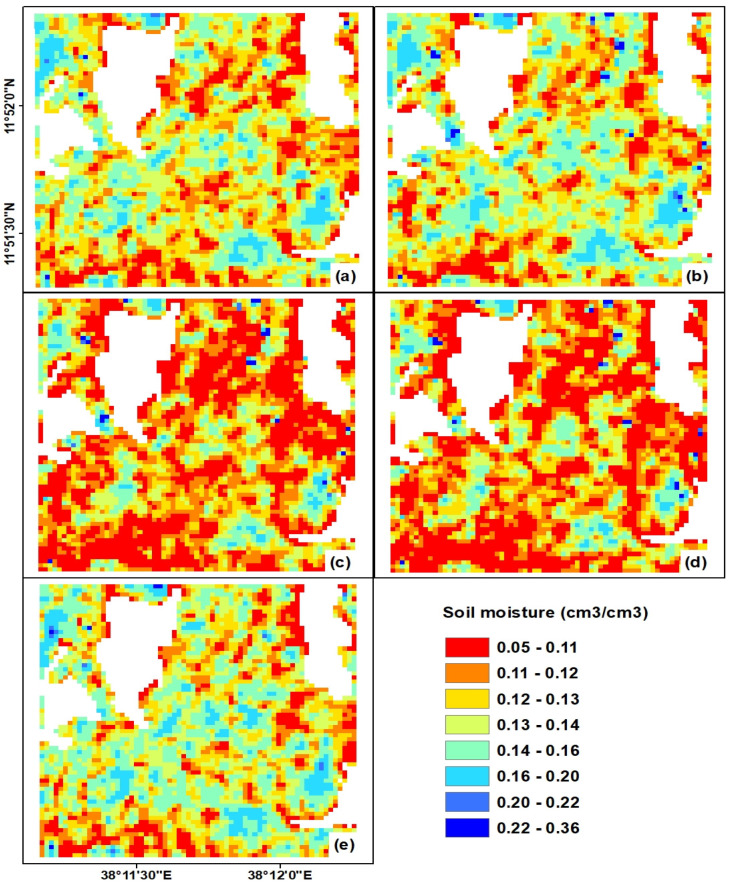
Spatio-temporal estimates of soil moisture based on the LRM method and $\sigma_{soil\_vv}$, $h_{rms}$, and $l_{eff}$ input variables for: (**a**) 22 November 2016, (**b**) 29 November 2016, (**c**) 16 December 2016, (**d**) 23 December 2016, and (**d**) 2 February 2017.

**Figure 11 sensors-20-03282-f011:**
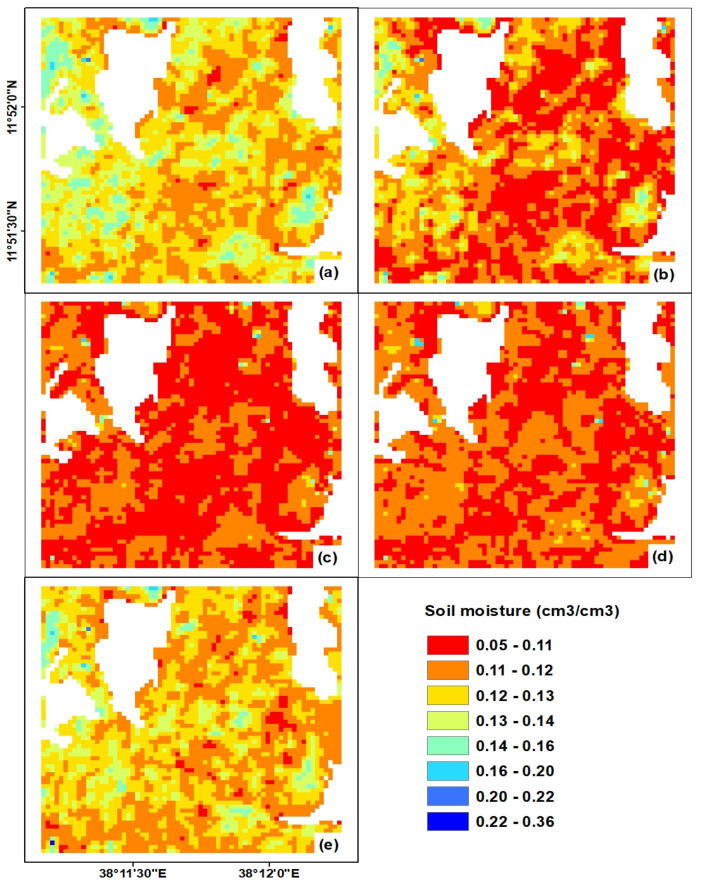
Spatio-temporal estimates of soil moisture based on the ANN method and $\sigma_{soil\_vv}$, $h_{rms}$, and $l_{eff}$ input variables for: (**a**) 22 November 2016, (**b**) 29 November 2016, (**c**) 16 December 2016, (**d**) 23 December 2016, and (**d**) 2 February 2017.

**Table 1 sensors-20-03282-t001:** Acquisition time, polarization, and incidence angle and orbit of Sentinel-1 Synthetic Aperture Radar (SAR) Interferometric Wide-Swath (IWS) Mode in the study area.

Date of Acquisition	Acquisition Time (UTC)		Polariz.	Incidence Angle	Orbit	Product Type
	Start	Stop				
22 November 2016	03:16:37	03:17:02	VV+VH	36.5°–39.0°	Descending	GRD
29 November 2016	15:34:57	15:35:22	VV	35.7°–38.7°	Ascending	GRD
16 December 2016	03:16:36	03:17:01	VV+VH	36.3°–38.9°	Descending	GRD
23 December 2016	15:34:56	15:35:21	VV	35.7°–38.6°	Ascending	GRD
2 February 2017	03:16:34	03:16:59	VV+VH	36.4°–39.0°	Descending	GRD

NB: VV represents vertical transmit and vertical receive polarization; VH for vertical transmit and horizontal receive polarization; and GRD represents Ground Range, Multi-Look, and Detected product type.

**Table 2 sensors-20-03282-t002:** Characteristics of Landsat images collected over the study site.

Date of Acquisition	Type	Sensor	Spectral Bands	Spatial Resolution (m)	Temporal Resolution (Day)
22 November 2016	Optical	Landsat 7	8	30	16
30 November 2016	Optical	Landsat 8	11	30	16
16 December 2016	Optical	Landsat 8	11	30	16
24 December 2016	Optical	Landsat 7	8	30	16
2 February 2017	Optical	Landsat 8	11	30	16

**Table 3 sensors-20-03282-t003:** The underlying vegetation parameters in a semi-empirical model.

Parameter	All Vegetation	Grazing Land	Crop	Grass
A	0.0012	0.0009	0.0018	0.0014
B	0.091	0.032	0.138	0.084

**Table 4 sensors-20-03282-t004:** Summary of the statistical performances of the soil moisture prediction model for the linear regression model (LRM) and the artificial neural network (ANN) methods using the three input configurations during the validation phase. The RMSE and MAE values are provided in terms of volumetric soil moisture (cm^3^/cm^3^).

Input Variables	LRM				ANN			
	MAE	RMSE	Bias	r	MAE	RMSE	Bias	r
$\sigma_{soil\_vv}$	0.030	0.040	−0.034	0.60	0.030	0.040	−0.032	0.57
$\sigma_{soil\_vv}$, $h_{rms}$	0.028	0.038	−0.019	0.70	0.028	0.036	0.000	0.67
$\sigma_{soil\_vv}$, $h_{rms}$, $l_{eff}$	0.027	0.037	0.014	0.70	0.026	0.035	−0.024	0.73

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
