# Peer review of "Combined Use of Sentinel-1 SAR and Landsat Sensors Products for Residual Soil Moisture Retrieval over Agricultural Fields in the Upper Blue Nile Basin, Ethiopia"

_sensors, 2020, doi:10.3390/s20113282_

Round 1
Reviewer 1 Report
1. There are some originality in the work since the authors are investigating a new study area, however, the novelty of the work is not highlighted, if there is.
2. What is the accuracy of the soil moisture prediction? How do you measure it? Please include it in the paper.
3. How accurate is the results compared to others work?
4. Since its using neural network for prediction, would the method applicable for other variants of data? E.g., other sensors data, other study area with different climate.
4. The authors should cross check the manuscript before submit it for review. There are some spotted formatting mistake such as, a) some figures are not easy to view, b) > 5 typos spotted, c) missing space between words, d) grammar mistakes.
Reviewer 2 Report
This paper aims to improve residual soil moisture monitoring by incorporating the surface roughness parameters into the prediction model with the help of ANN. The article seems to be very interesting. In general, the paper is well organized and the proposed methods are well described. However, the authors should improve the paper before publication in order to further validate the presented method in this paper.
- Several formulae are exploited in this paper. However, each formula is appeared in a sudden. For example, the reviewer wanders to know why the relationship ( Eq. (2) ) between VWC and NDWI is a quadratic function. One the one hand, the authors should present the cited references, where the formulae are from. One the other hand, the authors should simply describe the physical meaning about each formula. Based on these operations, the readers can easily understand each formula presented in the paper.
- Some symbols are not defined. For example, τ2 is not appeared in Eq.(3); however, the authors define a symbolτ2 in the paragraph below Eq. (3).
- In sections 2.3.2, the authors say that the interactions between vegetation and soil are neglected in the WCM. Generally, the authors should discuss the influence of interactions on the residual soil moisture monitoring. In my view, this influence can be clearly presented by comparing the residual soil moisture monitoring results based on (3) and (4). With this comparison, the consideration of interaction can be considered to be meaningful.
- Table 3 should be recreated as each title in the first line must be separated.
- Some figures such as Fig. 3, Fig. 6, and Fig. 9 are not clear. The authors should provide much more clear figures.
- Fig. 10 and Fig. 11 are generated by exploiting LRM method and ANN method, respectively. However, both results show major differences. The readers would be confused which result is close to the real one. Although quality parameters shown in Table 4 quantitatively indicate that the ANN method is superior to LRM method, there is no evidence to validate that the soil moisture shown in Fig. 11 is much more close to the real one. In order to evaluate the results, the standard result based on current optimum method should be produced at first.
Reviewer 3 Report
This paper investigates the potential of sentinel-1 SAR sensor products and the contribution of soil roughness parameters to estimate volumetric residual soil moisture in the UBN basin, Ethiopia. The ANN method was tested for its potential to translate SAR backscattering and surface roughness input variables to residual soil moisture values. The topic and approach of this study are very interesting. However, I have the following concerns which need the authors’ attentions before the paper can be considered to publish in this journal.
Comments:
- The Abstract of the work is suggested to be reduced properly, in order to make it highlight the contribution and main findings concisely.
- The novelty of the proposed algorithm is suggested to be further enriched in the introduction.?
- In the end of Section 1, the organization of this study is suggested to be summarized.
- Figure 6 and Figure 9 are not clear?
- What is the limitation of the proposed approach?
6. Some syntax errors or improper expressions exist, such as L3, L105, L167, L190, L228, L356, etc. Please double check the English.
Reviewer 4 Report
Dear Authors,
I made a maximum of comments on the pdf file. The article needs a lot of corrections.
Yours

Round 2
Reviewer 1 Report
I do agree to the authors that there is very limited research works on this topic in Ethiopia, but I still don't see the novelty of the research work. The presented work is another paper that applies published techniques into different study area, which uses machine learning to correlate the remote sensors data to the in-situ measurement data. The paper would be more interesting if the authors are able to find a new correlationship between the remote sensors data and the residual soil moisture, with a conclusion reached on the basis of evidence and reasoning.
Reviewer 2 Report
The paper is greatly improved after the revision. However, there are still some issues the authors should clearly addressed.
1.My privious comment that 'In sections 2.3.2, the authors say that the interactions between vegetation and soil are neglected in the WCM. Generally, the authors should discuss the influence of interactions on the residual soil moisture monitoring. In my view, this influence can be clearly presented by comparing the residual soil moisture monitoring results based on (3) and (4). With this comparison, the consideration of interaction can be considered to be meaningful' is not well addressed by the authors.
Generally, the authors consider the interaction between vegetation and soil in this paper. The precondition is that this interaction significantly affects the results. Therefore, I suggest that the comparing the residual soil moisture monitoring results based on (3) and (4) is conducted. This work would make the paper much more meaningful. However, the authors say that this interaction is not a dominating factor. This response is a contradiction. The authors should clarify this issue clearly.
2. Fig.3 and Fig. 4 are still not clear. The authors should provide much more clear figures.
Reviewer 4 Report
First of all I thank the authors for the responses sent to my comments and the changes made in the revised version of the paper. However, I do not share the conclusions of this study at all. In fact, a decrease of RMSE from 3.7 vol.% with LRM to 3.5 vol.% with ANN is not called "fairly well" but rather "similar performance" or "slightly improvment".
Similarly, between the three configurations tested, it is exactly the same conclusion. Indeed, a decrease of RMSE from 4 vol.% to 3.7 vol.% is not called improvement.
In general, an improvement is noted when the accuracy is improved by a factor of 2 or at the limit when there is an improvement of at least 30%. For example if the RMSE decrease from 6 vol.% to 4 vol% we can say improvement but if the RMSE decrease from 6 to 5 we say a slight improvement and when the RMSE decrease from 6 vol.% to 5.8 vol% then it is similar.
So the different conclusions in the paper must be redone: abstract, results, discussion and conclusion.
Response to comment 1: I am not convinced by your answer because a Sentinel-2 image acquired even 5 days after the Sentinel-1 image is also usable in your case. In fact, if your plots are of 30m x 30m, using landsat images at 30m resolution is not appropriate.
Response to comment 24: I do not agree with your answer. Your study site is small. The difference between the angles of incidence is around 3 °. A normalization will change the radar signal by 0.17dB (if I take incidences between 36 and 39 °). In this case and in your case, the normalization is not necessary. So, remove it from your paper. You can however add a sentence saying that for large areas and with very different angles it is necessary to normalize the radar signal.
Response to comment 49: add in the figure the equation of the linear regression because it is necessary to give the slope of the regression (= sensitivity)
Table 4: I think that the bias calculated for the last configuration in the case of LRM is not correct (it must be negative from Figure 9)
Response to comment 57: I am not convinced by your answer. In my opinion the bias is due to the simplified form of sigma_0 (Ok) But also it is due to the uncertainty of the models used.
Please verify the number of your refernces in the paper. For example I’m not sure that the reference [49] is good just before equation (9)
